# FcRn-Dependent Transcytosis of Monoclonal Antibody in Human Nasal Epithelial Cells In Vitro: A Prerequisite for a New Delivery Route for Therapy?

**DOI:** 10.3390/ijms20061379

**Published:** 2019-03-19

**Authors:** Emilie Bequignon, Christine Dhommée, Christelle Angely, Lucie Thomas, Mathieu Bottier, Estelle Escudier, Daniel Isabey, André Coste, Bruno Louis, Jean-François Papon, Valérie Gouilleux-Gruart

**Affiliations:** 1AP-HP, Hôpital Henri Mondor et Centre Hospitalier Intercommunal de Créteil, service d’Oto-Rhino-Laryngologie et de Chirurgie cervico-faciale, 94010 Créteil, France; andre.coste@chicreteil.fr; 2INSERM, U955, 94010 Créteil, France; christelleangely77@gmail.com (C.A.); bottier@wustl.edu (M.B.); daniel.isabey@inserm.fr (D.I.); bruno.louis@inserm.fr (B.L.); jean-francois.papon@aphp.fr (J.-F.P.); 3Université Paris-Est, Faculté de Médecine, F-94010 Créteil, France; 4CNRS, ERL 7240, 94010 Créteil, France; 5Université de Tours, EA 7501 GICC, F-37032 Tours, France; christine.dhommee@univ-tours.fr (C.D.); valerie.gouilleux@univ-tours.fr (V.G.-G.); 6Inserm U933, 75012 Paris, France; lucie.thomas@inserm.fr (L.T.); estelle.escudier@inserm.fr (E.E.); 7Université Pierre et Marie Curie, Faculté de Médecine, 75005 Paris, France; 8AP-HP Hôpital Armand-Trousseau, Service de génétique et d’embryologie médicale, 75012 Paris, France; 9AP-HP, Hôpital Bicêtre, Service d’Oto-Rhino-Laryngologie et de Chirurgie cervico-faciale, 94270 Le Kremlin-Bicêtre, France; 10Faculté de Médecine, Université Paris-Sud, F-94275 Le Kremlin-Bicêtre, France; 11CHRU de TOURS, Laboratoire d’Immunologie, F-37032 Tours, France

**Keywords:** neonatal Fc receptor, chronic rhinosinusitis with nasal polyps, transcytosis, human nasal epithelial cells, monoclonal antibodies

## Abstract

Monoclonal antibodies (mAbs) are promising therapies to treat airway chronic inflammatory disease (asthma or nasal polyps). To date, no study has specifically assessed, in vitro, the potential function of neonatal Fc receptor (FcRn) in IgG transcytosis through the human nasal airway epithelium. The objective of this study was to report the in vitro expression and function of FcRn in nasal human epithelium. FcRn expression was studied in an air–liquid interface (ALI) primary culture model of human nasal epithelial cells (HNEC) from polyps. FcRn expression was characterized by quantitative RT-PCR, western blot, and immunolabeling. The ability of HNECs to support mAb transcytosis via FcRn was assessed by transcytosis assay. This study demonstrates the expression of FcRn mRNA and protein in HNEC. We report a high expression of FcRn in the cytosol of ciliated, mucus, and basal cells by immunohistochemistry with a higher level of FcRn proteins in differentiated HNEC. We also proved in vitro transepithelial delivery of an IgG1 therapeutic mAb with a dose–response curve. This is the first time that FcRn expression and mAb transcytosis has been shown in a model of human nasal respiratory epithelium in vitro. This study is a prerequisite for FcRn-dependent nasal administration of mAbs.

## 1. Introduction

FcRn, or neonatal receptor for the Fc portion of IgG, was first identified in 1964 by Brambell [1]. It belongs to major histocompatibility complex (MHC) class I heterodimeric receptor family and consists of a type I transmembrane heavy chain that non-covalently associates with the soluble light chain, β2-microglobulin (β2m) [2]. In humans, FcRn was initially studied as a transporter of maternal IgG to the fetus. In adults, FcRn is also expressed in a wide variety of organs such as the lung, nose, skin, muscle, kidney, liver, and placenta. FcRn expression is detected in many cell types such as epithelial, endothelial, and hematopoietic cells [3]. In adults, studies have underlined the importance of FcRn in the regulation of serum IgG homeostasis [4,5,6]. The interaction of FcRn with its two ligands, IgG and albumin, is pH-dependent [7,8]. The binding of IgG and albumin to FcRn at acidic pH protects them from intracellular catabolism and thus prolongs the ligand half-life [7,9,10,11,12]. The FcRn recycling pathway is now well described. FcRn is responsible for IgG diversion from lysosomal degradation by an endosomal cellular recycling pathway after FcRn interaction [13]. IgG is taken into the cell by pinocytosis and processed within endosomes at a low pH environment that triggers the binding to FcRn. The routing of the endosomes transports IgG-FcRn complexes either by a transcytosis or recycling route dependent on the cell polarization [14]. A wide range of monoclonal antibodies (mAbs) have recently been developed to treat various cancers and inflammatory, autoimmune, allergic, or infectious diseases [15,16,17]. To further improve the use of this family of biopharmaceuticals, many parameters are currently being studied—choice of the target, FcRn-dependent and independent pharmacokinetics, and route of administration. While mAbs are usually administered intravenously, some are used subcutaneously [18,19]. Parenteral administration represents a limitation to their therapeutic use and has led to the development of other routes of administration. In mice and monkeys, the delivery of mAbs through the lower airways (aerosols were administered directly into the lungs through an endotracheal tube under general anesthesia) has been shown to be a promising development in the treatment of inflammatory respiratory diseases [20,21,22,23]. In line with these lung studies, Heidl et al. demonstrated the ex vivo expression of FcRn in nasal mucosa of inferior turbinate (fixed tissue) [24], and Samson et al. discussed FcRn-mediated transport through porcine nasal mucosa [25]. To date, no study has specifically assessed, in vitro, the potential function of FcRn in IgG transcytosis through the human nasal airway epithelium. The nasal route could be of great interest to pathologies affecting the upper airways, particularly for the treatment of chronic rhinosinusitis with nasal polyps (CRSwNP). The prevalence of CRSwNP ranges from 2% to 4% [26] and this disease is associated in 30% of patients with asthma and aspirin-exacerbated respiratory disease, a condition known as Samter’s triad [27]. In a recent review, Bachert reported a type 2 inflammatory pattern (involving expression of IL-4, -5, and -13 and increased concentrations of IgE) in 85% of patients with CRSwNP in western countries [28]. Some of these biomarkers are potential targets for innovative therapeutic approaches of CRSwNP, including mAbs directed against IgE (omalizumab), IL-5 (mepolizumab and reslizumab), and IL-4/IL-13 (dupilumab) [28,29,30]. MAbs have been tested subcutaneously or intravenously in proof-of-concept studies in patients with CRSwNP with or without asthma, and have demonstrated potential properties in reducing the volume of nasal polyps [28]. In the light of the results obtained in the lower airways, we hypothesized that nasal administration could be an interesting noninvasive alternative to the intravenous route to improve local mAb distribution and/or efficacy for treating CRSwNP. The objective was to evaluate the in vitro expression and function of FcRn in nasal epithelial cells. In the first part of this study, we evaluated the in vitro expression of FcRn in an air–liquid interface (ALI) model of primary culture of human nasal epithelial cells (HNEC) by quantitative RT-PCR, western blot, and immunolabeling. In the second part of the study, we evaluated the ability of HNEC to support mAb transcytosis.

## 2. Results

### 2.1. Expression of FcRn in Human Nasal Epithelial Cells (HNECs)

To investigate the possible use of the nasal route for mAb administration, we first evaluated FcRn expression in the HNEC culture, a well-described primary ALI model for nasal epithelium studies. The experiments were performed at the end of the first week (Day 7), when the cells were not yet differentiated, and at the beginning of the fourth week (Day 21) when the ciliated cells were fully differentiated (Figure 1).

RT-PCR performed in three primary cultures of HNECs showed an amplification of FcRn in the HNECs on both culture time day 7 and on day 21 (Figure 2A). Quantitative PCR showed that *FCGRT* mRNA levels isolated from two patient cells cultured for 21 days were significantly higher on day 7 compared to day 21 in two cell cultures (Figure 2B).

This statement is different to what is observed for FcRn protein expression by western blot. As detected by western blot, FcRn protein was expressed in the HNECs on both day 7 and day 21 culture times in all cultures but with a heterogeneity according to different cultures (no significant difference, *n* = 5 cultures). (Figure 3). Our findings revealed the presence of a large amount of FcRn protein in the fully differentiated cells compared to our positive control cell line (HEK 293) (×1.3 higher than in the HEK 293 control cell line) (Figure 3B). The discrepancies detected between mRNA and protein levels might be explained by their variable stability according to the different cell types and to the cellular differentiation status of the basal cells, goblet cells and ciliated cells.

### 2.2. Localization of FcRn in HNEC

We further studied the expression of FcRn by immunohistochemistry according to each cell type. We chose cytokeratin 14 staining to identify basal cells as it is one of the most intensely expressed proteins by this cell type [31]. MUC5AC staining was chosen to identify goblet cells as it is the first mucin expressed during in vitro mucus differentiation. Finally, acetylated α-tubulin was selected as a marker of microtubules in ciliated cells. Using these markers, we showed that FcRn is expressed in all epithelial cell types (Figure 4)—homogeneous distribution in the cytosol of basal and goblet cells, and no visible FcRn localization in the mucus vesicles. FcRn was predominantly localized in the apical part of the cytosol at the base of cilia and on the basolateral surface of ciliated cells.

### 2.3. In Vitro mAb Transcytosis in HNEC ALI Cultures

Once we had demonstrated FcRn expression in our HNEC culture model, we studied the possible transport of IgG mAbs across these polarized cells obtained from seven cultures. The results were expressed as absolute mAb quantities (12.5 to 1250 ng range) to compare apical-to-basal and basal-to-apical transcytosis that require different volumes of mAb solution (see methods section). We also compared mAb transfer according to the degree of HNEC differentiation at day 7 and day 21 (Figure 5). There was no significant difference between transepithelial electrical resistance (TEER) measurement before and after transcytosis essay (data not shown) and in all wells of experiments, no cellular toxicity was observed after a 4-h incubation period at 37 °C with infliximab.

The apical to basal experiment demonstrated effective mAb transfer across the HNECs. MAb transcytosis was detected on both day 7 and day 21 (no significant difference). The absolute quantities of transcytosed mAbs increased with mAb concentration loaded on the apical side with a mean of 3.25, 17.64, and 88.67 ng transcytosed mAb for 12.5, 125, and 1250 ng loaded mAb, respectively, on day 7 (Figure 5A). A basal-to-apical experiment was also performed with 375 ng of mAb in the well confirming mAb transcytosis toward the insert (Figure 5B). We calculated that three times more absolute mAb was required in the apical-to-basal direction compared to the basal-to-apical direction, in order to obtain an equivalent quantity of trancytosed mAbs in both directions.

Only cultures with a TEER value >400 Ω cm^2^ on day 7 and TEER values of ~600–800 Ω·cm^2^ on day 21 were selected. Values between T0 and 24 h later were not significantly different independently of differentiation status of culture (*p* > 0.4). The mean values of TEER on day 21 were 691 +/−64 Ω·cm^2^ before and 704 +/−33 Ω·cm^2^ after experiments. We had already published in vitro electrophysiological modifications in our ALI model of HNEC induced by aspergillus fumigatus. There was no difference after recovery for 24 h in ALI culture conditions, returning to values of T0 TEER in fungal filtrates and control wells [32]. According to another study of Lin et al. [33] ALI culture of human bronchial epithelial cell monolayers can be used as an in vitro model for airway drug transport studies when TEER values were higher than 500 Ω cm^2^. In their ALI model, when cultures are well differentiated, TEER values reached a relative stability over 6 days (around 600 Ω cm^2)^. In the literature, TEER values reflecting substantial barrier functions are typically >400 Ω cm^2^, with pneumacult-ALI medium [34]. In our ALI model with pneumacult-ALI medium, we already characterized TEER values and we observed a stability of values during one week with pneumacult-ALI medium.

## 3. Discussion

Monoclonal antibodies (mAbs) have become an important therapeutic option for several diseases. Innovations in antibody engineering have aimed at improving the pharmacokinetic properties of mAbs. The importance of FcRn in the pharmacokinetics of monoclonal antibodies is now well established [35]. In FcRn-knockout mice, mAbs have shown a plasma/serum half-life of only a few days, which demonstrates the importance of FcRn in their pharmacokinetics [36]. After nonspecific enodocytosis into cell, if IgG binds to FcRn in the endosome, the IgG–FcRn complex can be recycled back to the cell membrane with a release of antibody from endosome to plasma or transcytosis to opposite side.

Our primary ALI model of differentiated HNECs is very close to in vivo conditions [37]. It is, thus, a good tool for studying cellular and molecular mechanisms involving airway epithelial cells in many pathologies, and for evaluating toxic or pharmacological agents [38,39]. As nasal epithelium is constantly exposed to air, the ALI system is a polarized model (i.e., apical towards the air and basal towards the medium) that closely mimics the microenvironment and maintains the homeostasis of HNECs as in vivo. It can therefore be said that the model is highly suitable for the study of FcRn expression and mAb transcytosis to test our hypothesis.

In our study we were able to specify the expression of FcRn at the level of the nasal epithelium, at the level of each epithelial cell type, and we were able to highlight the efficiency of mAb transcytosis.

A number of studies have already shown FcRn expression in human epithelial cells, such as in the female genital tract, the lung, the intestinal tract, and in healthy nasal mucosa [20,24,40,41,42]. Altogether, our cultures of HNECs from NP confirm FcRn expression in vitro at both mRNA and protein levels. The discrepancies detected between mRNA and protein levels might be explained by their variable stability according to the various epithelial cell types and to the cellular differentiation status. Another hypothesis could be variable transcription and translation levels in each cell type. In the western blot assay, there was a difference in the density of beta-actin. Particularly, it appears that beta-actin expression, as determined by western blot, was different depending the differentiation state of HNEC. If beta-actin expression changes as HNEC reaches stable differentiated state, then beta-actin expression may not be a suitable normalizing protein control. It could also explain discrepancies between protein and mRNA. 

Similar FcRn cytoplasmic distribution has been described in human differentiated bronchial epithelial Calu-3 cell lines [43]. The apical localization of FcRn has previously been described in epithelial cells lining the rat intestine [44].

MAb IgG transcytosis has already been described across different epithelial cell lines (e.g., transfected rat kidney inner medullary collecting duct (IMCD), Madine–Darby canine kidney (MDCK), human colon epithelial cell lines T84, and CaCo-2) according the same experimental protocol [42,45] but it has never been studied in primary human epithelial cells cultures, especially nasal cells. In the same line, the IgG/FcRn or Fc-fused proteins/FcRn transport system is functional in several organs and tissues, such as the lung and intestine [23,46], but it has never been studied in the nasal mucosa.

In this study, we demonstrated, for the first time, that IgG transcytosis of mAbs is possible through an in vitro model of nasal epithelium, even in the presence of mucus. Infliximab was chosen as a mAb model due to our expertise in infliximab transcytosis in MDCKII cells transfected with human FcRn [45]. We tested a dose–response effect of mAb transcytosis in the apical-to-basal direction to mimic local administration of mAb in ALI cultures. The basolateral-to-apical transport was tested only in order to confirm data of literature about bidirectional transport of mAbs in other type of epithelial line cells. Basal-to-apical transport was found to be more efficient than apical-to-basal transport. This is in line with previous observations [42,47,48]. Indeed, compared to Foss’ in vitro cellular transport assay based on colon epithelial cell lines (T84) [42], we found that apical-to-basal mAb transcytosis required higher mAb concentrations. This could be explained by the presence of mucus on the apical surface in our model of HNEC ALI cultures which may represent a barrier for a mAb-based delivery system. In Foss’ study [42], they already demonstrated specific FcRn-mediated transcellular transport of IgG across polarized T84 cells grown on transwell using monoclonal human IgG1 Ab and a variant with mutations at the core of the FcRn interaction abrogating Ab binding to FcRn. In our study, we have not proved that transcytosis is Fc-mediated, but our results are consistent with that mechanism. In order to prove the rule of FcRn, the recent development of Efgartigimod (ARGX-113), a hIgG1-derived Fc fragment modified with ABDEG technology to block antibody recycling through FcRn binding, could be used in further experiments [49].

We performed our experiments at acidic pH since the ability of FcRn to protect IgG from intracellular catabolism and to transport them across the cells is the result of a specific pH-dependent interaction with the Fc portion of IgG at acidic pH. FcRn binds internalized IgG through pinocytosis in acidic endosomes, recycled it to the cell surface, and released it at neutral pH. At acidic pH, we observed a dose-dependent apical-to-basal transport of mAb. According to Foss, a rational design of Fc-engineered polymeric Fc-fusion molecules together with administration at mucosal sites containing an acidic environment may be an attractive approach for the efficient delivery of therapeutic molecules via FcRn [42]. A recent review underlines that a major process improving the pharmacokinetic properties of mAbs is FcRn-mediated recycling from endosome to plasma, which can be modulated by increased FcRn binding at acidic pH [35]. The nasal mucosa pH is of approximately 5.5–6.5 [50] and could thus represent an additional organ for the non-invasive delivery of therapeutic agents. Furthermore, nasal proteolytic activity is lower than in the intestine and the epithelium is naturally permeable to small peptides and proteins.

## 4. Methods

### 4.1. Primary Cultures of Human Nasal Epithelial Cells (HNEC)

HNECs were isolated from nasal polyps as previously described [51]. NPs were obtained from 20 patients with CRSwNP during ethmoidectomy. All the patients had given informed consent and the study was approved by the local ethics committee (CPP IDF X 2016-01-01). Briefly, the NPs were immediately placed in DMEM/F-12 supplemented with antibiotics (100 U/mL penicillin, 100 mg/mL streptomycin, 2.5 g/mL amphotericin B, and 100 mg/mL gentamicin) and transported to the laboratory for processing. Enzymatic digestion [0.1% (wt/vol) pronase in culture medium] was performed for 16 h at 4 °C. The HNECs (1 × 106 cells per well) were then plated in inserts (12-mm Transwell; Costar, MA, USA) with 12-mm-diameter polycarbonate micro-pore membranes (pore size of 0.4 µm) coated with type IV collagen (Sigma, Darmstadt, Germany) and incubated at 37 °C in 5% CO_2_. For the first 24 h, the cells were incubated with 1 mL of DMEM/F-12-antibiotics with 2% Ultroser G in the lower chamber and DMEM/F-12-antibiotics with 10% FCS in the insert. After 24 h, the culture medium (Stemcell, Pneumacult-ALI medium) in the insert was removed to place the cells at the ALI. The medium in the lower chamber was then changed every other day. The epithelial nature of the cultured cells has been already confirmed by flow cytometric analysis of cytokeratin immunofluorescent labeling showing 95% and 99% of positive cells on days 3 and 7, respectively [37]. The epithelial nature of the cultured cells has previously been well-demonstrated, leading to the use of primary culture of HNEC for several experiments aiming to study molecule expression [51,52,53]. The HNECs reached a stable differentiated state with the detection of ciliated, secretory, and basal cells during the third week of culture (Day 21) by using the ALI culture medium (Stemcell, Pneumacult -ALI medium) [54].

### 4.2. Quantitative Real-Time PCR (qRT-PCR)

Total RNA was isolated from primary cultures of HNECs at two culture times (day 7 and day 21) using the commercially available reagent QIAGEN kit. Human embryonic kidney cells (HEK-293) were used as positive control (ATCC, CRL-1573). Reverse transcriptase-polymerase chain reaction assays were performed using an RT kit Promega GO Script Reverse Transcription Detection System (Promega, ref A5000). Quantitative-PCR was carried out on the LightCycler 480 (Roche Diagnostics, Bâle, Switzerland). The level of *FCGRT* mRNA was normalized to the geometric mean of mRNAs of three reference genes: TATA-binding protein (*TBP*), 18S, and hypoxanthine phosphoribosyltransferase 1 (*HPRT1*). *FCGRT* expressions in HNEC were then reported to the expression level of HEK control cell line. Gene-specific primer pairs were designed according to published mRNA sequences. The sequences of the primers used in this study were: *FCGRT* F 5′-CCCTGGCTTTTCCGTGCTT-3′; R 5′-TGACGATTCCCACCACGAG-3′; *HPRT1* F 5′-CAT TATGCTGAGGATTTGGAAGG-3′; R 5′-CTTGAGCAC ACAGAGGGCTACA-3′; *TBP* F 5′-TGTATCCACAG TGAATCTTGGT TG-3′; R 5′-GGTTCGTGGCTCTCT TATCCTC-3′. Polymerase chain reaction was performed with PCR kit Promega GoTaq G2 Flexi DNA Polymerase M7805. PCR reactions were carried out using 2 with: 0 ng cDNA as a template, 0.2 μM each of forward and reverse primer, and 1× SYBR Premix Ex Taq (Takara Bio Inc, Kusatsu, Japan). Each reaction was performed in triplicate. The thermal protocol consisted of an initial denaturation step at 95 °C for 30 s followed by 40 cycles of denaturation at 95 °C for 5 s and primer annealing and extension at 60 °C for 20 s. Fluorescence was read at 60 °C during the annealing and extension step with an additional step (heating at 82 °C for 15 s) to record the fluorescence.

### 4.3. Western Blot

Primary Cultures of HNEC were homogenized in 200 µL RIPAbuffer (5 with: 0 mM Tris-HCl pH7.4, 15 with: 0 mM NaCl, with: 1 mM EDTA, 1% Na-desoxycholate, 0.1% sodium dodecyl sulfate (SDS), 1% NP-40)) containing protease inhibitors. Insoluble material was removed by centrifugation at 12,000× *g* for 10 min at 4 °C. Protein concentrations were determined using Bradford protein assay (Sigma-Aldrich, B6916, Darmstadt, Germany). Lysates of the HNEC cultures (20 µg) were separated on a NUPAGE 4–12% Bis-Tris gel (Life Technologies, Carlsbad, CA, USA) and then transferred onto a nitrocellulose membrane. The membranes were blocked with 5% non-fat milk in PBS and then incubated with mouse anti-FcRn Ab (1/500) (Santa Cruz Biotechnology, sc-271745). Anti-β-actin (1/5000) (Santa Cruz Biotechnology, sc-47778, Dallas, TX, USA) was used as a loading control antibody. After incubation with anti-mouse conjugated-HRP antibody, the membranes were developed using an enhanced chemiluminescence western blotting detection reagent (GE Healthcare, Chicago, IL, USA). HEK 293cell lysates were included as positive control. Densitometry of the bands was quantified using ImageJ. FcRn expression was normalized on β-actin and HNEC were then reported to the expression level of HEK control cell line.

### 4.4. Immunohistochemical Analysis

FcRn localization in HNEC cultures was characterized by immunolabeling using anti-FcRn (1/100) (Novus Biologicals cat. number NBP1-89128). Epithelial cells were collected by scraping the surface of the cell layers and cytocentrifuging them onto glass slides (700 rpm). They were then air dried and directly coimmunostained as described below. The cells were fixed with 4% paraformaldehyde for 15 min at room temperature, washed three times with PBS ++ (i.e., PBS supplemented with 0.4 with: 9 mM MgCl_2_ and 0. with: 9 mM CaCl_2_), incubated for 10 min with PBS++ NH_4_Cl 5with: 0 mM and then permeabilized with 0.1% Triton X-100 for 10 min. After rinsing twice with PBS++, the cells were incubated with rabbit polyclonal anti-FcRn antibody (Novus Biologicals cat. number NBP1-89128) (1 h) followed by an incubation with a goat anti-rabbit Alexa Fluor- 488 antibody (Invitrogen A11070) (1 h). They were then incubated either with mouse monoclonal anti-cytokeratin 14 (CellMarque 314M-14, 1/300) as a basal cell marker, mouse monoclonal anti-mucin-5AC (Abcam ab3649, 1/400) as a goblet cell marker, or mouse monoclonal anti-acetylated α-tubulin (Abcam ab24610, 1/700) as a ciliated cell marker, before being revealed by a secondary goat anti-mouse Alexa Fluor- 594 antibody (Invitrogen A11032). All the antibodies were incubated with 1% bovine serum albumin (Sigma-Aldrich, Darmstadt, Germany) in PBS++. The cells were finally washed three times with PBS++, mounted in ProLong^®^ Gold Antifade Reagent with DAPI (Cell Signaling #8961), and imaged on a Zeiss LSM 700 scanning laser confocal microscope (Carl Zeiss MicroImaging GmbH). Two negative controls were performed either by omitting the primary antibody or by using non-immune mouse serum.

### 4.5. Transcytosis Assay

We transposed a transcytosis protocol developed in human FcRn-transfected MDCKII cells using infliximab [45]. Infliximab was chosen as a mAb model due to our expertise in infliximab transcytosis in MDCKII cells transfected with human FcRn. The basal and apical sides of the inserts were washed twice with HBSS. Transepithelial electrical resistance (TEER) was monitored the day before the experiment using a MILLICELL-ERS-2 V-ohm meter (MILLIPORE) with a TEER value of ~600–800 Ω·cm^2^ to check the electrophysiological properties of the ALI cell cultures. In the literature, TEER values reflecting substantial barrier function are typically >400 Ω cm^2^, with Pneumacult-ALI medium [34]. Transcytosis experiments were performed either in the apical-to-basal direction or in the basal-to-apical direction. In the apical-to-basal transcytosis protocol, HBSS was buffered to pH 6.0 by the addition of MES buffer (Sigma-Aldrich, Darmstadt, Germany) in the upper chamber where the mAb was added, and to pH 7.4 by addition of HEPES buffer (Eurobio ingen, Les Ulis, France) in the lower chamber. After 30 min of incubation in these conditions, the volume of infliximab solution, added in the upper chamber, was 250 µL HBSS-MES (pH 6) for the apical-to-basal transcytosis. In the basal-to-apical transcytosis experiments, 750 µL of infliximab solution, was added in the lower chamber (pH 6). After a 4 h incubation period at 37 °C, as previously described [42], the medium was harvested (from either the basal or apical chamber, according to the experiment) to measure the infliximab concentration, using a validated ELISA assay [55]. Foss et al. [42] already reported in vitro experiments of transcytosis based on the human epithelial cell line T84. In this in vitro study, samples at 0, 1, 2, and 4 h were collected from the opposite side of the monolayer. The authors had already determined a time response effect with an optimal transcytosis time observed at 4 h. Therefore, we set the transcytosis time in our own experiments at 4 h. A dose–response curve was obtained by testing infliximab at concentrations of 50 µg/mL, 500 µg/mL, and 5000 µg/mL (representing 12.5, 125, and 1250 ng absolute values added in the upper chamber of well, respectively) in the apical-to-basal transcytosis experiments. In the basal-to-apical transcytosis experiments, the absolute value of 375 ng of infliximab (added in the lower chamber) was tested to confirm findings from previous studies. These ranges of concentrations were chosen according to previous studies [42]. Each mAb concentration was tested in triplicate. The experiments were performed in poorly differentiated (day 7) and fully differentiated (day 21) cultures to evaluate if the functionality of FcRn in HNEC was dependent on the cellular differentiation status. After each transcytosis experiment, we assessed cellular toxicity using both the TEER measurement for monolayer permeability and trypan blue exclusion for cellular viability. 

### 4.6. Statistical Analysis

The Mann-Whitney test was used to compare the quantitative values of FcRn expression (evaluated by qRT-PCR and western blot) and the absolute quantities of transcytosed mAbs between experiments performed on day 7 and on day 21. Statistical analysis was performed using the GraphPad Prism 5.0 (Inc., La Jolla, San Diego, CA, USA) software package. The alpha risk was set at 5%. The level of significance is indicated in the figures as * for a *p* ≤ 0.05, ns: non-significant.

### 4.7. Ethics Approval and Consent to Participate

All subjects gave their informed consent for inclusion before they participated in the study. The study was conducted in accordance with the Declaration of Helsinki, and the protocol was approved by the local ethics committee (CPP IDF X 2016-01-01).

## 5. Conclusions

In this study, we report in vitro FcRn expression in HNEC in a differentiated model of ALI primary culture and demonstrate that FcRn expression is dependent on the degree of cell differentiation. Our major finding is that we demonstrate in our model a transepithelial passage of therapeutic IgG mAb with a dose-dependent response curve. This is the first time that FcRn expression and mAb transcytosis has been shown in a primary culture of human nasal cells from NP. From a drug-delivery perspective, this study is a prerequisite for FcRn-dependent nasal administration of therapeutic mAbs. Further experiments are required to study this new local pathway with innovative mAb treatments for CRSwNP (i.e., omalizumab, mepolizumab, reslizumab, and dupilumab). In addition, new mAb formats such as antibody-drug conjugates coupled to glucocorticoids [56] could also be tested to increase the effectiveness of an established treatment standard for CRSwNP as well as for other nasal diseases such as allergic rhinitis which affects 400 million people throughout the world.

## Figures and Tables

**Figure 1 ijms-20-01379-f001:**
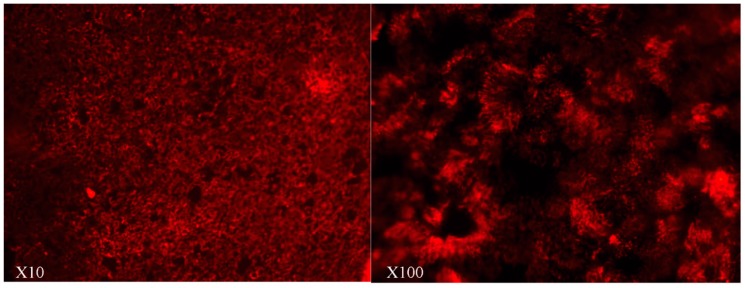
In vitro differentiation of human nasal epithelial cell in air–liquid interface culture. Immunostaining in transwell inserts after three weeks of culture (Day 21) using the anti-βIV tubulin monoclonal antibody (1/500) (Abcam ab-11315). Cilia of a differentiated epithelium are shown in red. (Confocal images: ×10 (left image) and ×100 (right image)): ciliated structures were clearly detected on the apical surface of the cells.

**Figure 2 ijms-20-01379-f002:**
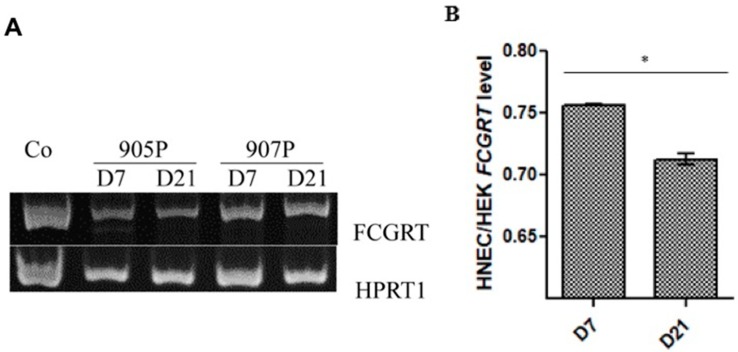
FCGRT mRNA expression in air–liquid interface culture of human nasal epithelial cells. RT-PCR with HPRT1 used as housekeeping gene (**A**) and qRT- PCR (**B**) showing amplification of FCGRT mRNA isolated from five different cultures (905P, 907P, and 895P for PCR and 954P and 956P for quantitative PCR) of human nasal epithelial cells (HNEC) on day 7 (D7) and day 21 (D21). HEK 293 was included as positive control (Co). FCGRT expression in HNEC were then reported to the expression level of HEK control cell line. Results are presented as mean +/− SEM. * *p* < 0.05.

**Figure 3 ijms-20-01379-f003:**
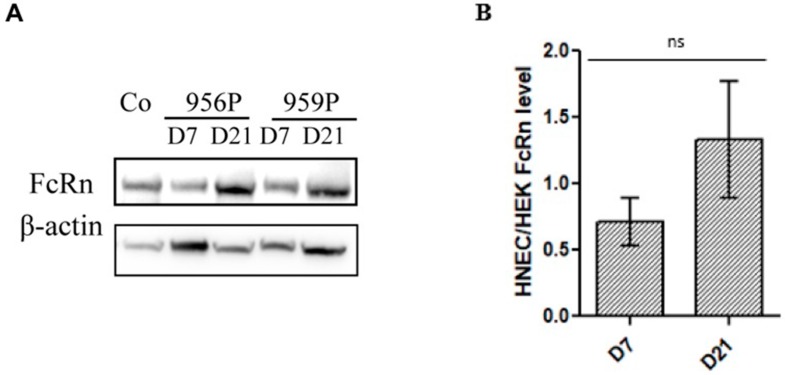
Neonatal Fc receptor (FcRn) protein expression in air–liquid interface culture of human nasal epithelial cells. (**A**) Western Blot analysis of FcRn expression using anti-FcRn antibody (1/500, Santa Cruz Biotechnology, sc-271745) in HNEC on day 7 (D7) and day 21 (D21) in two out of five representative different cultures (956P and 959P). HEK 293 cell extract was included as positive control (Co). β-actin was used as control for equal protein loading. (**B**) FcRn expression was normalized on β-actin. FcRn expression in the five cultures of HNEC were then reported to the expression level of HEK control cell line. Results are presented as mean +/− SEM. * *p* < 0.05.

**Figure 4 ijms-20-01379-f004:**
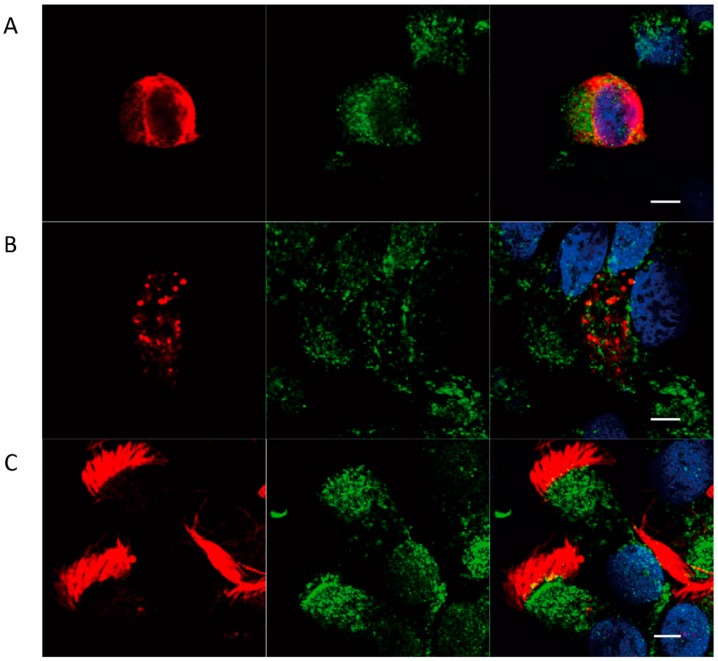
FcRn localization in air–liquid interface culture of human nasal epithelial cells on day 21. Cells were co-immunolabeled for FcRn (green) and (**A**) a basal cell marker, cytokeratin-14 (red), or (**B**) a goblet cell marker, mucin-5AC (red), or (**C**) a ciliated cell marker, acetylated α-tubulin (red); and imaged by confocal microscopy. White scale bars represent 5 μm.

**Figure 5 ijms-20-01379-f005:**
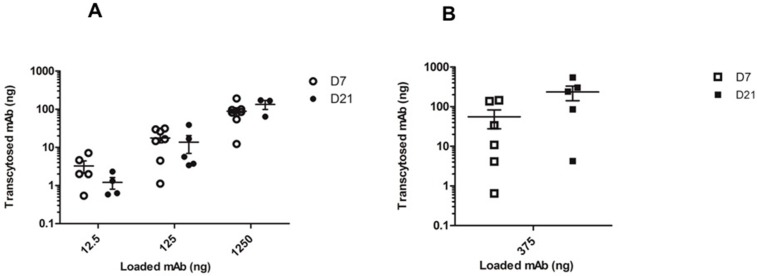
Monoclonal antibody (mAb) transcytosis through air–liquid interface culture of human nasal epithelial cells at day 7 and day 21 of different cell cultures (*n* = 7) (954P, 955P, 956P, 959P, 994P, 997P, and 1002P). Transcytosis was tested in (**A**) apical-to-basal transport (on day 7—solid circles; on day 21— open circles) or (**B**) basolateral-to-apical transport (on day 7—solid squares; day 21—open squares). Results are expressed in absolute quantity of mAb in each compartment. Bars represent means +/− SEM of the different experiments.

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
