# Peer review of "FcRn-Dependent Transcytosis of Monoclonal Antibody in Human Nasal Epithelial Cells In Vitro: A Prerequisite for a New Delivery Route for Therapy?"

_ijms, 2019, doi:10.3390/ijms20061379_

Round 1
Reviewer 1 Report
General Comments
The authors demonstrated the FcRn mRNA and protein expression by human nasal epithelial cells (HNEC) at two differentiated states. In addition, the authors demonstrated the localization of FcRn by immunohistochemistry. The authors also demonstrated the bidirectional transcytosis of the IgG mAb transcytosis by HNEC. This study provides the evidence of the presence of FcRn in nasal epithelial cells and its function in IgG transcytosis, which may serve as a base for further study on FcRn-dependent nasal administration of mAbs. This article is neatly-written, and the contents are logically presented. The results are generally supportive of the conclusions, though more clarifications are needed. This article is recommended to be published with minor revisions.
Specific Comments
1. “The binding of IgG and albumin to FcRn at acidic pH protects them from intracellular catabolism and thus prolongs the ligand half-life.”
a. The mechanism of the prolonged half-life is not well explained. A more detailed explanation of FcRn recycling pathway is recommended.
2. I have several comment related to the assessment of FcRn mRNA and protein expression:
a. The authors stated that FCGRT mRNA level is higher in patient cells cultured for 21 days compared to patient’s cells cultured for 7 days (Line 112-115). However, Figure 2B appears to demonstrate the opposite, where HNEC/HEK FCGRT level at day 7 is higher than at day 21. In the subsequent Western blot analyses, both the authors’ statement (Line 123-128) and Figure 3B point to higher FcRn protein expression at day 21 compared to day 7. Please clarify why the authors’ statement comparing FcRn mRNA expression level at day and day 21 appears to contradict Figure 2B.
b. In the western blot assay, there appears to be significant differences in the density of beta-actin. Particularly, it appears that beta-actin expression, as determined by Western Blot, is different depending the differentiation state of HNEC. If beta-actin expression changes as HNEC reaches stable differentiated state, then beta-actin expression may not be a suitable normalizing protein control. Have the authors performed replicate Western Blot analyses to demonstrate constant beta-actin density across different samples?
3. The authors state: “We calculated that 3 times more absolute mAb was required in the apical to basal direction compared to the basal to apical direction, in order to obtain an equivalent quantity of trancytosed mAbs in both directions.” (Line 165-167) The authors later states “…we found that apical to basal mAb transcytosis required higher mAb concentrations. This could be explained by the presence of mucus on the apical surface in our model of HNEC ALI cultures which may represent a barrier for mAb-based delivery system.” (Line 217-220)
a. In order to justify the authors’ claim, have the authors attempted to experimentally determine whether the observed difference in efficiency in mAb transcytosis (apical to basal vs. basal to apical) is the result of the presence of mucus at apical side, or the result of FcRn acting as an efflux transporter of IgG from basal to apical side.
Minor Comments
1. “In mice and monkeys, the delivery of mAbs through the lower airways (aerosols were administered directly into the lungs through an endotracheal tube under general anaesthesia). has been shown to be a promising development in the treatment of inflammatory respiratory diseases.” (Line 74-77)
a. There is an extra period after the right parenthesis.
2. “A recent review underlines that a major process improving the properties of mAbs. is FcRn-mediated recycling from endosome to plasma, which can be modulated by increased FcRn binding at acidic pH [30].”
a. There is an extra period after in this sentence.
Author Response
Response to Reviewer 1 Comments
I thank you for taking the time to consider our manuscript and for your pertinent comments
“The binding of IgG and albumin to FcRn at acidic pH protects them from intracellular catabolism and thus prolongs the ligand half-life [1-5]. The FcRn recycling pathway is now well described. FcRn is responsible for IgG diversion from lysosomal degradation by an endosomal cellular recycling pathway after FcRn interaction [6]. IgG is taken into the cell by pinocytosis and processed within endosomes at a low pH environment that triggers the binding to FcRn. The routing of the endosomes transports IgG-FcRn complexes either by a transcytosis or recycling route dependent on the cell polarization [7].”
This paragraph as well as two references have been added in the manuscript (lines 68-73, ref 13 and 14):
13.Roopenian, D. C., Christianson, G. J., Sproule, T. J., Brown, A. C., Akilesh, S., Jung, N., Petkova, S., Avanessian, L., Choi, E. Y., Shaffer, D. J., Eden, P. A., and Anderson, C. L. (2003) The MHC class I-like IgG receptor controls perinatal IgG transport, IgG homeostasis, and fate of IgG-Fccoupled drugs. J. Immunol. 170, 3528–3533
14.Ward, E. S., Zhou, J., Ghetie, V., and Ober, R. J. (2003) Evidence to support the cellular mechanism involved in serum IgG homeostasis in humans. Int. Immunol. 15, 187–195
Point 2: I have several comment related to the assessment of FcRn mRNA and protein expression:
2a: The authors stated that FCGRT mRNA level is higher in patient cells cultured for 21 days compared to patient’s cells cultured for 7 days (Line 112-115). However, Figure 2B appears to demonstrate the opposite, where HNEC/HEK FCGRT level at day 7 is higher than at day 21. In the subsequent Western blot analyses, both the authors’ statement (Line 123-128) and Figure 3B point to higher FcRn protein expression at day 21 compared to day 7. Please clarify why the authors’ statement comparing FcRn mRNA expression level at day and day 21 appears to contradict Figure 2B.
Response 2a: Lines 120-122, we stated that “Quantitative PCR showed that FCGRT mRNA levels isolated from 2 patient cells were higher in patient cells cultured for 7 days compared to same patient’s cells cultured for 21 days (Figure 2B).”
This statement is different to what is observed for FcRn protein expression by western blot. As detected by western blot, FcRn protein was expressed in the HNECs on both day 7 and day 21 culture times in all cultures but with an heterogeneity according to different cultures (no significant difference, n=5 cultures). (Figure 3). Our findings revealed the presence of a large amount of FcRn protein in the fully differentiated cells compared to our positive control cell line (HEK 293) (x1.3 higher than in the HEK 293 control cell line) (Figure 3B). The discrepancies detected between mRNA and protein levels might be explained by their variable stability according to the different cell types and to the cellular differentiation status of the basal cells, goblet cells and ciliated cells.
2b- In the western blot assay, there appears to be significant differences in the density of beta-actin. Particularly, it appears that beta-actin expression, as determined by Western Blot, is different depending the differentiation state of HNEC. If beta-actin expression changes as HNEC reaches stable differentiated state, then beta-actin expression may not be a suitable normalizing protein control. Have the authors performed replicate Western Blot analyses to demonstrate constant beta-actin density across different samples?
Response 2b: We agree with your comment. We added your comment in the discussion (lines 209- 213) because this could also explain the discrepancies between protein and mRNA.
Point 3: The authors state: “We calculated that 3 times more absolute mAb was required in the apical to basal direction compared to the basal to apical direction, in order to obtain an equivalent quantity of trancytosed mAbs in both directions.” (Line 165-167) The authors later states “…we found that apical to basal mAb transcytosis required higher mAb concentrations. This could be explained by the presence of mucus on the apical surface in our model of HNEC ALI cultures which may represent a barrier for mAb-based delivery system.” (Line 217-220)
a. In order to justify the authors’ claim, have the authors attempted to experimentally determine whether the observed difference in efficiency in mAb transcytosis (apical to basal vs. basal to apical) is the result of the presence of mucus at apical side, or the result of FcRn acting as an efflux transporter of IgG from basal to apical side.
Response 3c: In some experiments (data not shown) we removed excess mucus from the apical surface by washing the cells once per week with D-PBS (without Ca++ and Mg++) at room temperature (15 - 25°C) as proposed in the protocol of Stemcell to prevent excessive mucus accumulation and one day before experiments. We didn’t find a difference in efficiency in mAb transcytosis (apical to basal vs. basal to apical) between experiments with or without removal of mucus.
Minor Comments
Point 4 : “In mice and monkeys, the delivery of mAbs through the lower airways (aerosols were administered directly into the lungs through an endotracheal tube under general anaesthesia). has been shown to be a promising development in the treatment of inflammatory respiratory diseases.” (Line 74-77)
There is an extra period after the right parenthesis.
Response 4: We agree with your comment. We removed the extra period after the right parenthesis
“In mice and monkeys, the delivery of mAbs through the lower airways (aerosols were administered directly into the lungs through an endotracheal tube under general anaesthesia) has been shown to be a promising development in the treatment of inflammatory respiratory diseases [8-11].”
Point 5 : “A recent review underlines that a major process improving the properties of mAbs. is FcRn-mediated recycling from endosome to plasma, which can be modulated by increased FcRn binding at acidic pH [30].”
a. There is an extra period after in this sentence.
Response 5: We agree with your comment. We removed the extra period after the right parenthesis (lines 257)
“A recent review underlines that a major process improving the properties of mAbs is FcRn-mediated recycling from endosome to plasma, which can be modulated by increased FcRn binding at acidic pH [30].”
Reviewer 2 Report
The paper of Bequignon, entitled “FcRn-dependent transcytosis of monoclonal antibody in human nasal epithelial cells in vitro: a pre-requisite for a new delivery route for therapy?” discusses transcytosis of an IgG1 monoclonal antibody through human nasal epithelial cells using a primary culture model. There are some issues that may form a threat to the validity of the conclusions drawn in this paper.
1. For instance, the authors measured the TEER value of the culture one day before the experiment and therefore assume that the TEER does not change anymore. However, the authors do no provide evidence or a reference for this assumption. It is quiet likely that in such a primary culture TEER values are dynamic and thus may not be the same 24 hours later
2. Next the authors performed the transcytosis experiment and deduced that transcytosis occurred after detecting antibodies in the opposite chamber. However, a crucial experiment where transcytosis is blocked (e.g., with drugs or anti-FcRn) has not been conducted. In the light of the concern described above, such an experiment must be included to conclude something about trancytosis.
3. Several models of transcytosis have shown that it is a relatively fast process. It would have been more informative if the authors had looked at the kinetics of trancytosis, by measuring the amount of monoclonal antibody transcytosed every 30 minutes for instance.
Author Response
Response to Reviewer 2 Comments
I thank you for taking the time to consider our manuscript and for your pertinent comments
Comments and Suggestions for Authors
The paper of Bequignon, entitled “FcRn-dependent transcytosis of monoclonal antibody in human nasal epithelial cells in vitro: a pre-requisite for a new delivery route for therapy?” discusses transcytosis of an IgG1 monoclonal antibody through human nasal epithelial cells using a primary culture model. There are some issues that may form a threat to the validity of the conclusions drawn in this paper.
POINT 1: For instance, the authors measured the TEER value of the culture one day before the experiment and therefore assume that the TEER does not change anymore. However, the authors do no provide evidence or a reference for this assumption. It is quiet likely that in such a primary culture TEER values are dynamic and thus may not be the same 24 hours later
Response 1: We are agree with your suggestion to add some references about stability of TEER values in ALI model of epithelial cells and data from our study before and after experiments:
“Only cultures with a TEER value >400 Ω‐cm2 on day 7 and TEER values of ~600–800 Ω·cm2 on day 21 were selected. Values between T0 and 24 hours later were not significantly different independently of differentiation status of culture (p>0.4). The mean values of TEER on day 21 were 691 +/-64 Ω·cm2 before and 704 +/-33 Ω·cm2 after experiments. We had already published in vitro electrophysiological modifications in our ALI model of Human nasal epithelial cells induced by aspergillus fumigatus. There was no difference after recovery for 24 hours in air-liquid culture conditions, returning to values of T0 TEER in fungal filtrates and control wells [12]. According another study of Lin et Al [13]. Air-liquid interface (ALI) culture of human bronchial epithelial cell monolayers can be used as an in vitro model for airway drug transport studies when TEER values were higher than 500 Ω‐cm2. In their ALI model, when cultures are well differentiated, TEER values reached a relative stability during 6 days (around 600 Ω‐cm2). In literature, TEER values reflecting substantial barrier function are typically >400 Ω‐cm2, with Pneumacult-ALI medium [14]. In our ALI model with pneumacult-ALI medium, we already characterized TEER values and we observed a stability of values during one week with Pneumacult-ALI medium. “
This paragraph has been added in the “Transcytosis assay “ result section (lines 184-198)
POINT 2: Next the authors performed the transcytosis experiment and deduced that transcytosis occurred after detecting antibodies in the opposite chamber. However, a crucial experiment where transcytosis is blocked (e.g., with drugs or anti-FcRn) has not been conducted. In the light of the concern described above, such an experiment must be included to conclude something about trancytosis.
Response 2: We are agree with this suggestion: inhibition of the IgG recycling activity of FcRn is a valid strategy to prove the rule of FcRn. The recent development of efgartigimod (ARGX-113), a hIgG1-derived Fc fragment modified with ABDEG technology, could block antibody recycling through FcRn binding. The development work on ARGX-113 is done in close collaboration with Prof. E. Sally Ward (University of Texas Southwestern Medical and Texas A&M University Health Science Center, a part of Texas A&M University (TAMHSC)). Efgartigimod has recently completed phase I clinical trial and phase II in myasthenia gravis but was not available outside of clinical trials in France. We plan to use ergartigimod in further experiments in order to inhibit the FcRn-dependent IgG transcytosis.
However, we first need to obtain ARGX-113 from S. Ward and then perform inhibition experiments in primary cultures of human nasal epithelial cells. This will take at least 4 months to perform these new sets of experiments and include it in our paper. That’s why we ask you if this experiment is absolutely necessary in order to meet the criteria of publication of IJMS and ask for extra delay in this case.
POINT 3: Several models of transcytosis have shown that it is a relatively fast process. It would have been more informative if the authors had looked at the kinetics of trancytosis, by measuring the amount of monoclonal antibody transcytosed every 30 minutes for instance.
Foss et Al. already reported in vitro experiments of transcytosis based on the human epithelial cell line T84. In this in vitro study, samples at 0, 1, 2 and 4 h for initial experiments and after 0 and 4 h for remaining experiments were collected from the opposite side of the monolayer. The authors had already determined a time response effect with an optimal transcytosis time observed at 4H. Therefore we set the transcytosis time in our own experiments at 4H. We added this reference that helped us to determine our protocol in methods section (lines 355-359). In order to respond to this point we could also perform kinetics of transcytosis but we need extra delay in this case.
We added this reference: Foss S, Grevys A, Sand KM, Bern M, Blundell P, Michaelsen TE, et al. Enhanced FcRn-dependent transepithelial delivery of IgG by Fc-engineering and polymerization. J Control Release. 2016;223:42-52.
Round 2
Reviewer 2 Report
The authors replied to most of my comments. Although I do not think that solid conclusions can be drawn without proper elimination experiments, the results described in this paper provide some good level of evidence that would allow other scientist to look more carefully into the subject and design proper elimination experiments.
Author Response
I thank you for taking the time to consider our manuscript and for your pertinent comments. We agree we have not proved transcytosis is Fc-mediated and we added the following paragraph in the discussion section:
" In S. Foss’ study [42] they already demonstrated specific FcRn-mediated transcellular transport of IgG across polarized T84 cells grown on transwell using monoclonal human IgG1 Ab and a variant with mutations at the core of the FcRn interaction abrogating Ab binding to FcRn. In our study, we have not proved transcytosis is Fc-mediated but our results are consistent with that mechanism. In order to prove the rule of FcRn, the recent development of Efgartigimod (ARGX-113), a hIgG1-derived Fc fragment modified with ABDEG technology to block antibody recycling through FcRn binding could be used in further experiments."